# A Study on the Design of Bending Process According to the Shape of Initial Billets for Bi-Metal Elbow

**Seonghun Ha, Daehwan Cho** **, Joonhong Park and Seongjae Kim \***

Department of Mechanical Engineering, Dong-A University, 37, Nakdong-Daero 550 Beon-gil, Saha-gu, Busan 49315, Korea
\* Correspondence: sjkim641@dau.ac.kr; Tel.: +82-051-200-7647

**Abstract:** Studies have been steadily conducted on the forming process of the bending pipe that enables the transport of underground resources. Recently, it has been suggested that bent pipes for transport withstand high pressure during the forming process, but it is judged that the research on methods able to overcome the limitations of non-uniform dimensional distribution due to the difference in the mechanical properties and thickness of the outer and inner pipes is insufficient. This study proposes a new precision forging method called the cut-forged-joint process (CFJP) for the manufacture of bent pipe containing bi-metal. The initial billet and mandrel were designed considering the standard dimensions of bent pipes, and pre-simulation was performed applied to the designed models. The results of dimensional accuracy obtained by forging experiments and the computational forming simulation were compared with each other to verify the reliability. As a final outcome, it was confirmed that it is possible to secure the dimensional accuracy of bi-metal bent pipes by applying the newly proposed CFJP.

**Keywords:** bi-metal; cut-forged-joint process (CFJP); material loss rate; precision forming analysis; short radius elbow (SR); uniform dimensional distribution

## 1. Introduction

Land resource reserves continue to decrease as the consumption of natural resources increases in various fields. The mining environment of natural resources has gradually expanded from land to abyss or severely cold regions. Therefore, there are continuously increasing demands for special alloy pipes to be adopted in transport steel pipes and marine structures [1]. Among such pipes, lined and clad pipes suitable for high-pressure environments with corrosion and heat resistance have been the focus of active research. These pipes are special alloy pipes with excellent corrosion resistance made by bonding different materials, i.e., a corrosion-resistant alloy (CRA) and steel for the inner and outer pipes, respectively. The difference between the two pipes lies in the method of bonding their constituent materials. Lined pipes use mechanical bonding, while clad pipes adopt metallurgical bonding [2]. The former is applicable to relatively more fields, as its manufacturing equipment is less expensive than that of the latter and because it enables lightweight design [3].

The hydroforming process is used to fabricate lined pipes using a die with specific geometry, while the outer pipe is mechanically bonded with the inner pipe, which is subjected to plastic deformation by applying hydraulic pressure using its elastic recovery force. However, for the bonding method via the hydroforming process, bonding occurs only within the elastic section of the outer pipe material. It is also difficult to constantly apply high hydraulic pressure to the inner side during the production of long pipes or pipes with a large diameter because this may decrease the bonding strength between the different materials. The application of the cold drawing process can be considered instead of the hydroforming process to address the limitations of the bonding strength during

the production of lined pipes. This method facilitates the application of high pressure to each pipe in a relatively uniform manner. Studies have also been actively conducted on producing curved pipes by applying the bending forming process to the lined pipes fabricated to construct a transport system. Christopher et al. conducted research on the method of optimizing the design of simplified tool geometry based on the contact normal stress distribution of dies according to the bending radius of the curved pipe in the rotary draw bending process (RDBP) [4]. Andrea et al. proposed an analytical approach to predict the stress–strain distribution of the pipe cross section, including the spring-back phenomenon as thermodynamic process parameters in the RDBP [5]. Safdarian investigated the effects of the RDBP parameters on the fracture, wrinkling, and ovality of piping via experiments and numerical analysis [6]. Ruan et al. examined the applicability of the hydroforming process, considering the RDBP using numerical and experimental methods to produce an ultra-small bending radius elbow with STS304 [7]. Kong et al. studied the deformation behavior and the failure in bending-bulging forming which is good at fabricating elbow parts with small relative bending radius [8].

In addition, research on the direct extrusion bending process has been actively conducted in recent years. Shiraishi et al. predicted the curvatures of the rectangular bars and tubes with flanges by combining the fundamental bending properties of the process obtained in extrusion of the rectangular bars with the shape effect of the bars and tubes [9]. Zhou et al. considered an analytical model for quantitatively describing the bending behavior of aluminum profiles produced in a novel extrusion process and proposed a novel method for directly forming curved profiles/sections from billets in one extrusion operation using two opposing punches [10,11].

For lined pipes, the inner pipe, made of CRA, is designed to have a smaller wall thickness than the outer pipe made of steel, although its strength is generally lower. Although the inner pipe produced using the RBDP has usually lower strength than the outer pipe made of steel in lined pipes, it is designed to have a smaller wall thickness. Hence, lined pipes have inherent issues, such as the buckling and wrinkling that occur in the inner pipe with relatively weaker strength, the separation of the inner and outer pipes during the metal forming due to the thickness difference, and the disparity between the centers of the bending radii of the pipes after forming.

This study proposes the cut-forging-joint process (CFJP), a novel bi-metal curved pipe forming method that introduced the forging process to minimize the difference in thickness between the processed curved pipes. Although the existing RDBP performs forming in order of bending–cutting–joining, the CFJP is performed in order of cutting–forging–joining. This approach can minimize the amount of discarded material generated by cutting a part after bending the manufactured curved pipe in the RDBP and secure the final geometry with a uniform wall thickness [12]. It can also control the load and set the machining allowance through the initial material to perform forging, including the design of the mandrel's dimensions [13]. In this study, the CFJP was applied as a method for manufacturing double-curved pipes, while the formability and dimensional precision were analyzed according to the process parameters to secure the reliability of the process design. Figure 1 shows a schematic diagram of the overall process procedure. The initial material and geometry of the die suitable for the manufacturing of targeted short radius elbows (SR) in accordance with the curved pipe standards were designed with appropriateness based on the dimensional precision of the final prototype by performing finite element simulation beforehand [14,15]. The possibility of forming bi-metal curved pipes within the tolerance of standards was verified via comparison with the prototype fabricated using the actual double curved pipe forming experiment.

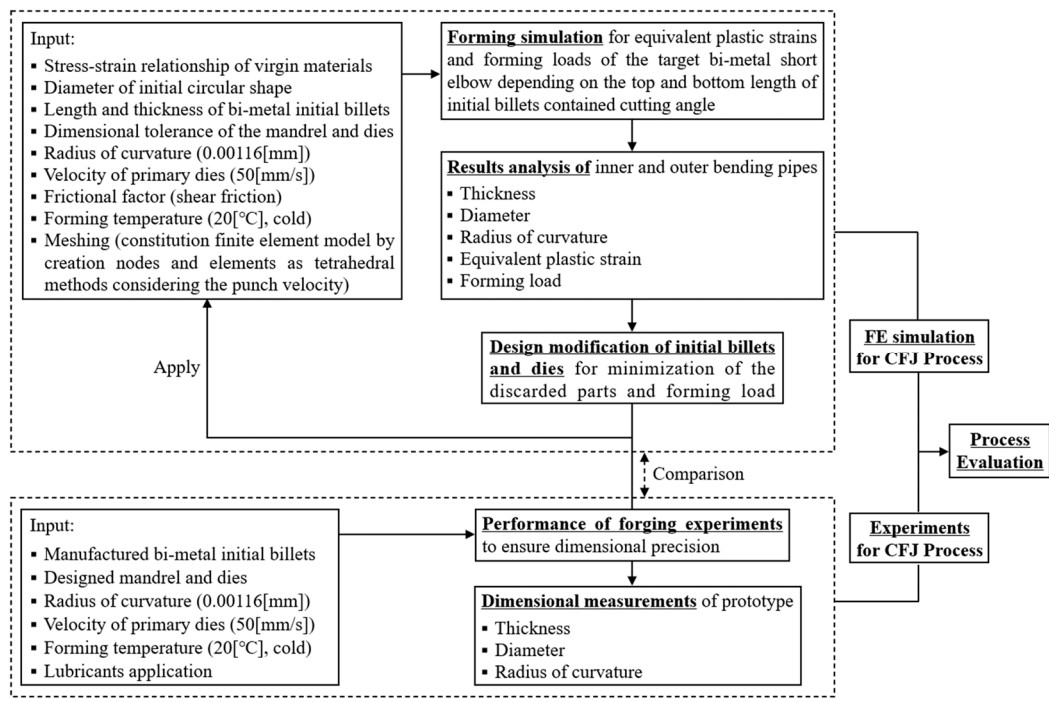

**Figure 1.** Design procedures for CFJP evaluation.

## 2. Materials and Methods

### 2.1. Materials

The outer pipe is primarily made of carbon steel, such as API 5L X65 (Dongyeun, Busan, Korea), which can withstand high pressure during the manufacturing of lined pipes. For the inner pipe, duplex stainless steel (DSS, Dongyeun, Busan, Korea), which has the benefits of ferritic and austenitic stainless steel, is mainly employed [16]. API 5L X65 grade pipes are mainly used for supplying natural gas in Korea. They have high yield strength and are primarily employed to produce large pipes. DSS, a steel containing both ferrite and austenite by adding nickel to ferritic stainless chromium-based steel, is suitable for use in environments where stress corrosion cracking may occur, owing to its high tensile strength, weldability, and corrosion resistance with a low thermal expansion coefficient. Thus, in this study, API 5L X65 carbon steel and DSS were selected shown in Table 1 as materials for the outer and inner pipes respectively. For the adopted DSS, the iron, nickel, chromium, and molybdenum contents—which are elements that improve the local corrosion resistance of DSS—were higher than those of other elements [17].

**Table 1.** Chemical composition of applied materials (API 5L X65, Duplex).

| Materials | Chemical Composition, wt. [%] | | | | | | | | | |
|---|---|---|---|---|---|---|---|---|---|---|
| | C | Mn | P | S | Si | Cr | Ni | Mo | N | Fe |
| API 5L X65 | 0.16 | 1.45 | 0.019 | 0.01 | - | - | - | - | - | Bal. |
| Duplex | 0.03 | 2.0 | 0.045 | 0.03 | 0.75 | 18.0 | 14.0 | 3.0 | 0.1 | Bal. |

### 2.2. Methods

#### 2.2.1. CFJ Process

In the forging process, steel is heated to a certain temperature and pressurized to form the desired shape. Forging is divided into free and die forging. For free forging, which does not use a separate die, relatively little energy is required for forming; however, the work speed is slow, and its dimensional precision is low. Die forging is a method of

forming by applying external force with a forging machine after placing steel in a die. It is one of the most difficult manufacturing processes to design. Nevertheless, die forging can secure high strength and dimensional precision while rapidly producing products with complex geometry, implying that the introduction of die forging to the entire forming process can be considered to secure excellent quality and precise dimensions during the manufacturing of lined pipes using different materials [18,19]. In order to produce curved pipes that comply with the target dimensional specifications in this study, actual forging tests with this additional work were required. However, the reliability of the forming load and formability that occurs depending on the shape of the initial billets composed of bi-metal during the forging process has not yet been verified, and an in-depth analysis is required. A novel dimensioning method that can minimize the geometry deviation of the double short radius elbow using the different materials formed via the CFJ process in Figure 2 was suggested by conducting finite element analysis using designated billets, and its reliability was verified through comparison with the dimension measurement results after the forming actual products through forging machine.

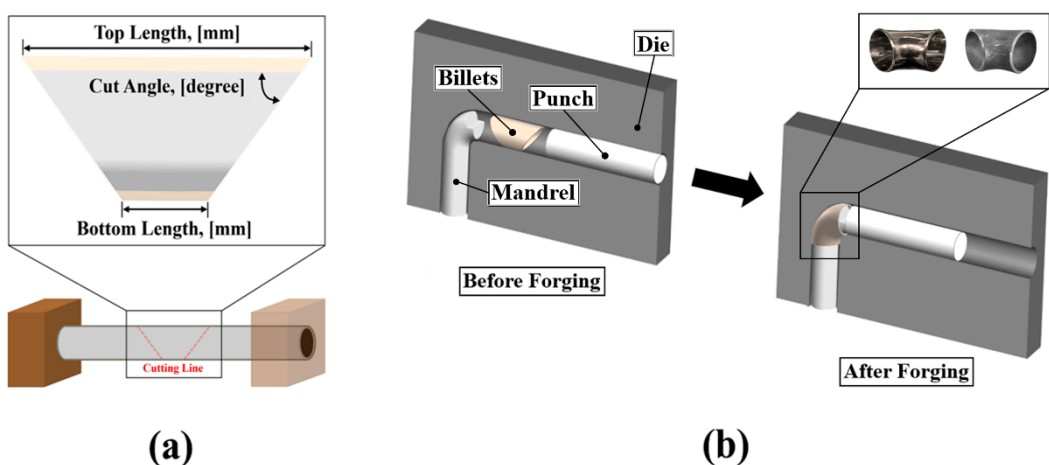

**Figure 2.** Schematic illustration of the CFJ process: (**a**) cutting process; (**b**) forging–bending process.

2.2.2. Design of Initial Billet and Die

To determine the design of the dimensions of the initial material, including the necessity of the mandrel for the cutting process—performed before the forging process in the CFJ process described above—the influence of the geometry of the initial material cut at a specific angle on both sides on the geometry of the final curved pipe prototype produced via forging was analyzed beforehand. In addition, forming simulation was also performed beforehand to establish the selection criteria for the main process parameters, such as the geometry of the mandrel and the initial material cutting angle. DEFORM Ver 12.0 was used as forming software. For the reference billet model to be applied to the initial forming analysis, the target SR was assumed to be a sector with a central angle of 90°. Based on the characteristic that the product of the radius and the central angle is equal to the length of the arc, the minimum values of the top and bottom lengths could be assumed to be 190 mm and 50 mm, respectively. When the law of constant volume during forming was considered with the inference that the target SR and initial material volumes are 138,540.15 mm$^3$ and 139,259.63 mm$^3$ for top and bottom lengths of 190 mm and 50 mm, respectively, it was determined that the new dimensional design considering the volume change after cutting the formed product would be required.

For double pipes, more deformations may occur in the material with relatively lower strength due to their different mechanical properties. Considering the additional considerations for the weld zone, the initial material design cannot be performed with completely identical standards as single pipes. Therefore, it is necessary to analyze the difference in the geometry of the forged product depending on the design parameters of the initial material.

The machining allowance at the bottom was set to 10 mm for face milling to be applied after the forging process, and the marginal range of the ratio between the overall length and bottom length was determined to be between 0.3 and 0.5 to minimize the amount of discarded material. The dimensions of the initial material based on the initially set target SR and forming analysis results are presented in Figure 3 and Table 2. These values do not include the machining allowance after forming. These are the optimum values based on the initial billets with a top length of 200 mm and a bottom length of 60 mm and the measurement of the loss rate according to the secondary cutting length value. In this regard, as a result of designing the shape parameters and analyzing them through pre-forming simulation, it was confirmed that the total cut length was reduced by up to 8–9 mm when a vertical cut length of 15–16 mm was applied.

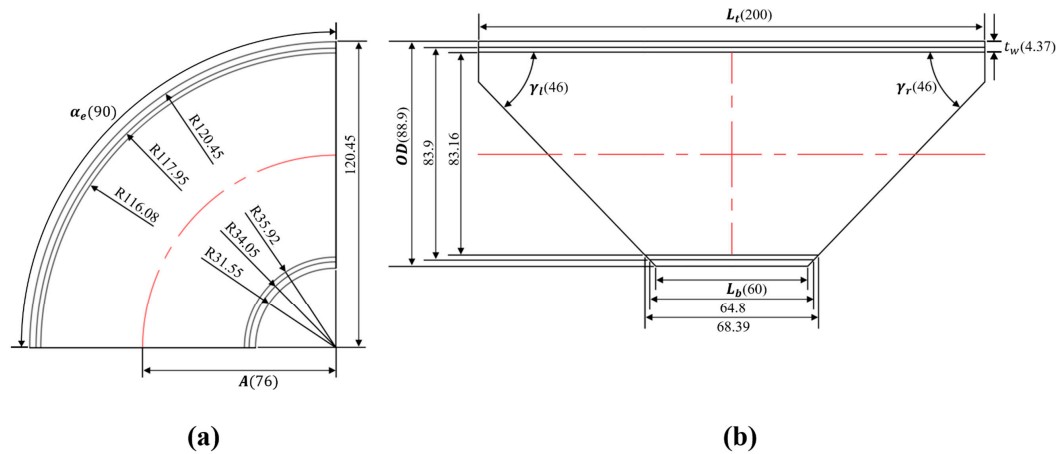

(a)          (b)

**Figure 3.** Geometry of initial billet and die: (**a**) target short radius elbow; (**b**) designed initial billet.

**Table 2.** Design parameter of target SR and billet.

| Models | Parameters | Symbols | Values | Units |
|---|---|---|---|---|
| Target SR | Diameter nominal | $DN$ | 80 | - |
| | Outside diameter | $OD$ | 88.9 | mm |
| | Specified wall thickness | $t_w$ | 4.37 | mm |
| | Center to end | $A$ | 76 | mm |
| | Angle of elbows | $\alpha_e$ | 90 | degree |
| Initial Billet | Top length | $L_t$ | 200 | mm |
| | Bottom length | $L_b$ | 60 | mm |
| | Left cutting angle | $\gamma_l$ | 46 | degree |
| | Right cutting angle | $\gamma_r$ | 46 | degree |

2.2.3. Formability Evaluation of Bi-Metal Pipe using Forging Analysis

In this study, the top length was fixed at 200 mm and the bottom length ranged from 60–100 mm such that the ratio of the bottom length to the top length of the initial material could range from 30–50%, and the initial material model was designed for each case. The outside diameter and wall thickness of the forged product were calculated by conducting a forging analysis to evaluate the roundness after forming. The objective here is to select the proper initial material and die geometry that can reduce the prototype fabrication cost in designing CFJP for producing curved pipes in precise dimensions. To calculate the load applied to the initial material and its geometry change, finite element modeling (meshing) comprised tetrahedral grids with 17,535 nodes and 62,975 elements for the outer pipe and 13,790 nodes and 41,420 elements for the inner pipe. Although the die, mandrel, and punch are rigid bodies, the raw material to be formed was set as a deformable body.

Considering that separation may occur owing to the bonding characteristics of lined pipes, cold shear friction conditions—which are mainly used in the analysis of the friction

behavior of solids with a large aspect ratio, such as pipe materials—were applied. The differences in curved pipe formability depending on the presence or absence of sticking were analyzed. Regarding the contact friction coefficient, the friction coefficient value of the lubricant mainly used in the press forging process (0.02) was applied to the contact surface between the die and mandrel, while 0.9 was assigned between the bi-metal and punch considering continuous pressurization. For the elastic moduli of API 5L X65 and DSS, the correction curve was extended by assuming the plastic section during large deformation in the true stress–true strain curve; the true stress values (209 GPa and 199 GPa) at a true strain of 1 were applied. In addition, Poisson's ratios were set to 0.3 and 0.32, respectively.

In the plastic section, the overall flow curve for large deformation was adopted by applying the Hollomon equation to the basic property data obtained via the tensile test; Table 3 presents the detailed property data applied to each material. It is advantageous to predict the ideal work and forming load for each process via calculation through calculus up to the true fracture strain of the flow curve obtained through the uniaxial tensile test. This is based on the effective stress–effective strain function, which was previously based on the second-order invariant function of von Mises. Regardless of the general stress axis system or the principal stress system, the total value of effective stress is always the same, and when considering the three-dimensional stress tensor in the case of a uniaxial tensile test, the element values for shear stress other than the normal stress acting in the tensile direction are 0. This means that even if the abnormality is calculated based on the load and stress in the uniaxial direction secured by the tensile test, the abnormality in the non-uniaxial complex stress state is also calculated as the same value.

**Table 3.** Mechanical properties of inside and outside billet.

| Materials | Mechanical Properties | | | | |
|---|---|---|---|---|---|
| | Yield Strength [MPa] | Tensile Strength [MPa] | Strength Coefficient [MPa] | Strain Hardening Exponent [-] | Elongation [%] |
| API 5L X65 | 476 | 660 | 711 | 0.087 | 28 |
| Duplex | 250 | 550 | 1300 | 0.512 | 35 |

The press speed of the punch for forging double-curved pipes was set to 50 mm/s. Here, 3D cold forging was selected as the forming type. Regarding the material characteristics, the rigid plastic solution with high accuracy and calculation efficiency considering geometry implementation and bulk forming analysis was applied. Boundary conditions in three axial directions were not separately assigned to consider the rotation triggered by the moment that may emerge on the material during forming. For the fracture condition, the normalized Cockcroft and Latham equation was employed. It was assumed that softening occurred at 95% of the fracture strain based on the actual property data. The forming temperature and convective heat transfer coefficient were set to 20 °C and 0.02, respectively. The conjugate gradient method was set for the solver, with the maximum number of iterations set to 200 by applying the Newton–Raphson iterative method such that the error rate could converge to less than 5%.

### 2.2.4. Experimental Measurement of Bent Pipes

As the pre-work for the initial material design, an actual forging experiment was performed to obtain the precise thickness measurement value of the formed curved pipe as shown on Figure 4. For the initial material model adopted in the experiment, the top length was fixed at 200 mm and the bottom length was designed in the 60–100 mm range such that the value of LR could be between 30–50%. This range was selected considering top and bottom lengths of 200 mm and 60 mm, respectively, as minimum length conditions based on the pre-computation and simulation results. This is the standard measured by using the cut angle according to the lower length setting as a design variable in a state

where secondary cutting of the initial billets does not occur. In addition, the initial material geometry case adopted in the experiment was selected considering the amount of discarded material during the face milling of the forged product produced via the bending process. API 5L X65 and DSS were employed for the outer and inner pipes, respectively. For the thickness of the initial material, an experimental model with a thickness of 2.5 mm for the outer pipe and 1.87 mm for the inner pipe was adopted. If the tolerance inside the die is reduced by increasing the diameter of the mandrel, the amount of increase in thickness decreases, and it is possible to secure a shape close to the target size, but the load will increase in turn. This means that in the optimization criterion of 50 to 190, an additional material length is needed to increase the thickness instead of lowering the forming load. The maximum allowable tolerance for the thickness of the target, 45° short radius elbow, is 1.6 mm. Although the main purpose of this study is to produce curved bi-metal pipes with a uniform thickness, the experiment was conducted except for designing the maximum or minimum tolerance through the mandrel to prevent excessive load during the forging process. A mandrel with a tolerance of ±0.8 was installed as a response to the increase in load by the friction force between the material and the mandrel. The prototype was fabricated according to the cut initial material geometry using a press speed of 50 mm/s, with the dimensions of each formed product measured after face milling.

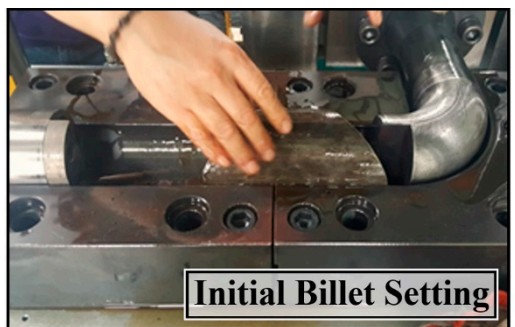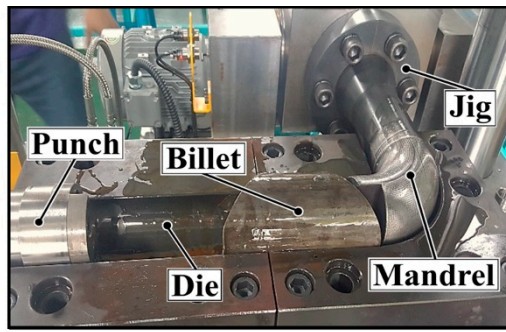

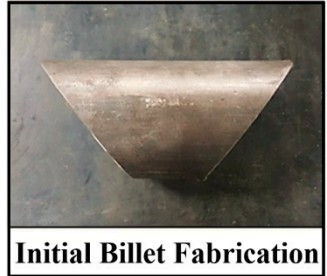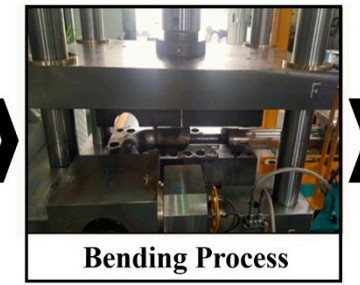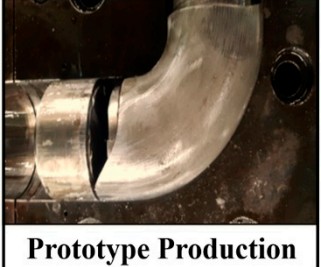

**Figure 4.** Fabrication process of short radius elbow.

## 3. Results

Relative to the forging simulation results, the deformation geometry depending on the setting of forming conditions was secured, as illustrated in Figure 5. Figure 5a presents the deformation of the single pipe condition made of the DSS without mandrel. Compared to the result of using the cut lined pipe as the initial billets, it was shown that the overall thickness of Figure 5a increased more than Figure 5b. The measured average thicknesses of the top and bottom parts were 4.51 mm and 6.32 mm, respectively, on the single pipe. As bending deformation occurred at the forming point, the thickness of the top end tended to increase more than other parts owing to the load acting at the top increasing by up to 6.38 mm. The central part of the bottom thickness was also bent more than necessary owing to insufficient internal pressure, and the thickness increased by up to 6.63 mm. Compared to the target overall length, the bottom length exceeded 53.74 mm for the single pipe and 54.27 mm for the double pipe, thereby verifying that the design requires the bottom part of the initial material to be cut off.

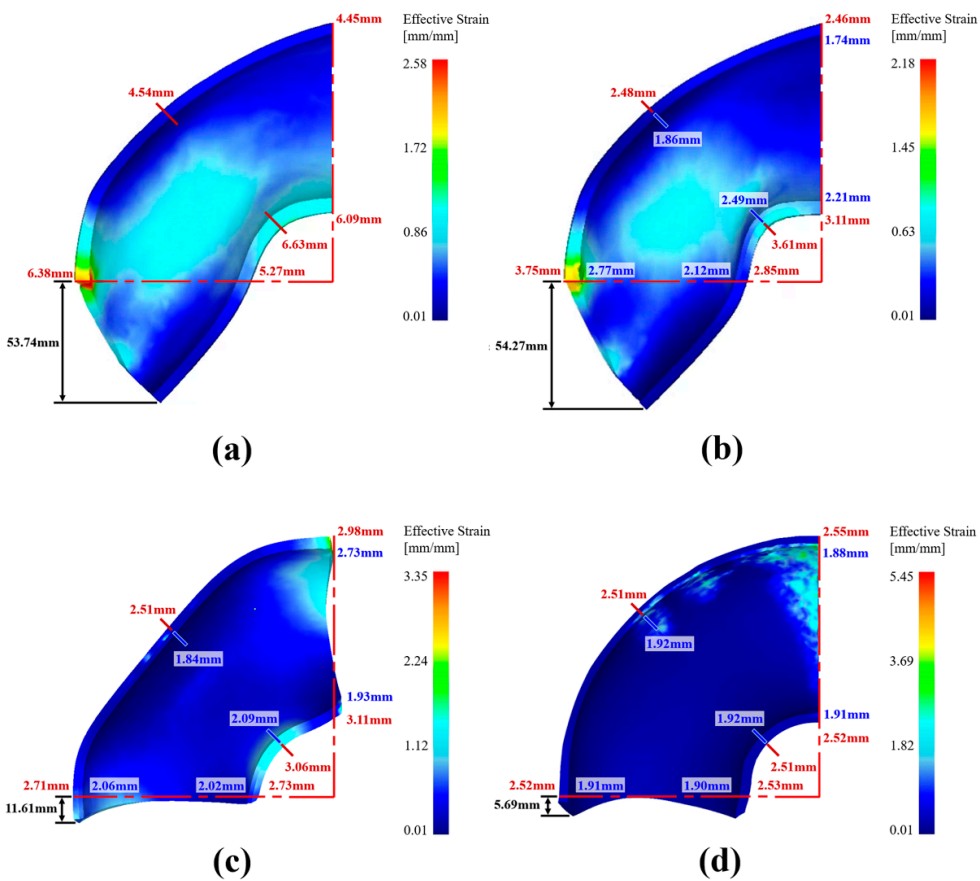

**Figure 5.** Predicted shape of elbow using forging analysis contained experiments. (**a**) the deformation of the single pipe condition made of the DSS without mandrel; (**b**) the deformation of bi-metal pipes condition; the deformation geometry depending on the setting of other forming conditions (**c**,**d**).

The deformation of bi-metal pipes condition shown in Figure 5b comprising the outer pipe made of APIX 5L X65 and the inner pipe made of DSS without a mandrel. The average top thicknesses were calculated to be 2.89 mm and 2.12 mm for the outer and inner pipes, respectively, while the average bottom thicknesses were 3.19 mm and 2.27 mm. These values demonstrate that the thickness sensitivity is higher for the forming of the lined pipe than the single pipe owing to the difference in mechanical properties between the outer and inner pipes. The maximum forming load of single pipe and lined pipe was calculated to be 45.8 tons and 50.7 tons, respectively, based on when the punch moved by 190 mm, when mandrel is used, it is thought the load will rise further due to frictional force than before. These results imply that the forming load significantly impacts the thickness accuracy of the curved pipe, making it necessary to apply a method of securing dimensional precision via the appropriate dispersion of the load. Therefore, the necessity of designing and introducing the mandrel for the type of each pipe models was also confirmed to secure the roundness of the formed product, minimize the thickness change of the top and bottom parts, and obtain uniform thickness distribution.

The bottom geometry of the formed products according to the cutting angle of the initial billets demonstrated that the tip of the top-right part was bent by up to 5.8 mm to the left when pressurized with the punch. While the right part was not cut, deformation on the top right part occurred more as the length ratio decreased. In addition, for the left part of the initial material, the geometry of the bottom left side was curved upward as forming progressed. Finally, a cutting length of at least 13.1 mm was required from the bottom-left part to the cutting limit line. In addition, the required cutting line increased proportionally as the length ratio increased. These results indicate that the initial billets must be designed

with the consideration that the bottom part of the final prototype contained the machining allowance that enables a cut of at least 13.1 mm. In addition, its top right part should allow a cut of 5.8 mm when pressurized, regardless of the right part's geometry after designating the cutting angle on the left side. These are the results obtained based on the billets that do not involve cutting on the right side. The values must be modified if the cutting of the right side is assumed. When the machining allowance of up to 20 mm is considered for top and bottom lengths of 190 mm and 50 mm, respectively—which are dimensions calculated using the geometrical method to secure stable machining allowance and produce a curved pipe with the target dimensions—it is possible to consider 190–210 mm and 50–70 mm for the top and bottom lengths, respectively. When the top length of the initial billets was set to 200 mm, its bottom length and cutting angles on both sides were set as design variables with the main purpose of material loss minimization according to the objective of this study. The bottom length that can minimize the required cutting length was determined to be 60–70 mm, as shown in Figure 6 and Table 4. These values were considered assuming that the maximum loss length is 5 mm during additional face milling after the forging process.

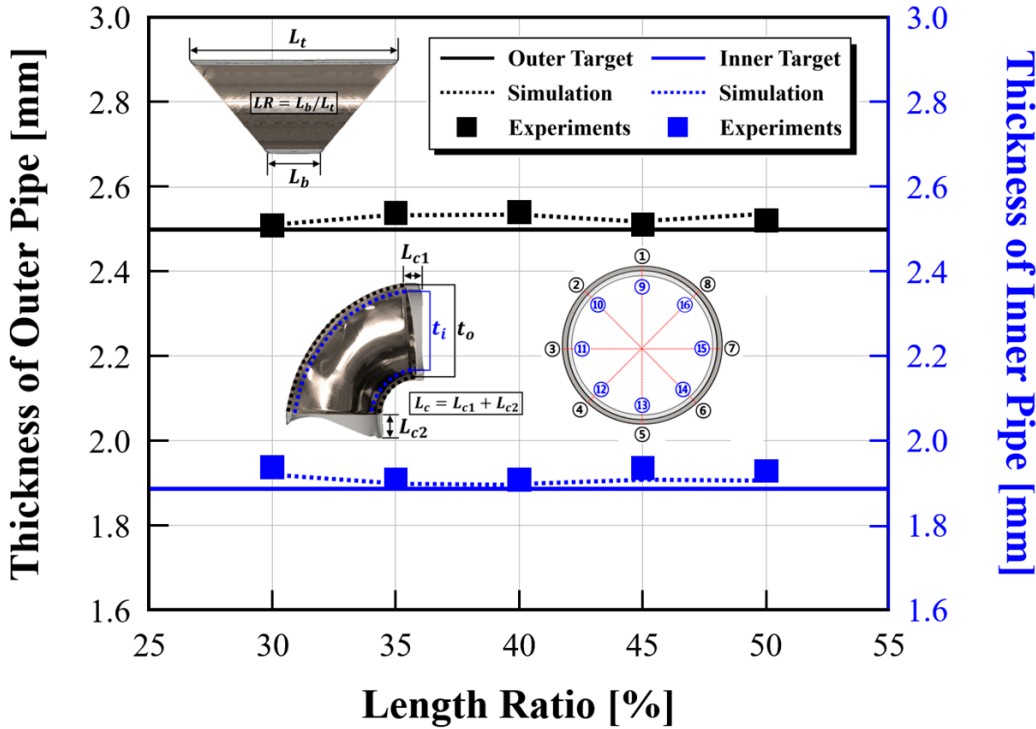

**Figure 6.** Results of dimensional precision measurement according to length ratio.

The minimum forming load was calculated to be 87 tons at a bottom length of 60 mm, while the maximum forming load was 97 tons at a bottom length of 100 mm. Consequently, appropriate mandrel geometry was selected for controlling the loads acting on the top and bottom parts, and the proper forming process design for bi-metal curved pipe was realized using the top and bottom lengths and tolerances. In all designed cases, the outside diameter measured by simulation was the equal to 88.9 mm, and it was confirmed that the error rate was calculated to be within 0.22%, compared to the average diameter of the manufactured product that had undergone the actual forging test, which was 89.1 mm. The overall average thickness of the product was 4.45 mm for the experimental values and was 4.429 mm for the analysis values. The overall thickness showed a tendency to increase after forging compared to the initial billets. For the same top length, the change rate of the outer thickness according to the bottom length did not show a specific trend, which means that the standard for predicting the thickness of the outer and inner pipe cannot be determined solely by the top and bottom length of the initial material. The

difference in thickness change between the experimental and the analysis values using forming simulation program is thought to be caused by the fact that the actual inner space of dies is wider than the space of the finite element model used in the forming simulation.

**Table 4.** Dimensional measured values of forged bi-metal billet.

| Symbol | $L_t$ | $L_b$ | $t_o$ | $t_i$ | $L_c$ | LR |
|---|---|---|---|---|---|---|
| Units | mm | mm | mm | mm | mm | % |
| | 200 | 60 | 2.514 | 1.915 | 16.17 | 30.0 |
| | 200 | 70 | 2.532 | 1.897 | 21.64 | 35.0 |
| Simulation | 200 | 80 | 2.538 | 1.891 | 27.17 | 40.0 |
| | 200 | 90 | 2.524 | 1.905 | 32.82 | 45.0 |
| | 200 | 100 | 2.528 | 1.901 | 38.67 | 50.0 |
| | 200 | 60 | 2.512 | 1.938 | 15.61 | 30.0 |
| | 200 | 70 | 2.539 | 1.908 | 21.04 | 35.0 |
| Experiments | 200 | 80 | 2.541 | 1.911 | 26.56 | 40.0 |
| | 200 | 90 | 2.515 | 1.937 | 32.14 | 45.0 |
| | 200 | 100 | 2.523 | 1.926 | 38.02 | 50.0 |

By calculating the volume of the bi-metal curved pipe with increased thickness through the forging process and considering the volume of material loss caused by cutting, processing allowance and tolerance setting range can be determined. The total thickness during forming increases similarly overall for the initial billets designed in this study. Thus, it is assumed this is the standard loss volume that occurs primarily. In addition, the required cut length generated depending on the design shape of the initial billets and the sum of the secondary loss volume and the primary loss volume should be considered together. However, additional cutting related to the primary and secondary loss volume may not be necessary in some cases according to shape results. If it is arbitrarily assumed that about half of the primary loss volume at the bottom end of the manufactured product is lost in a symmetrical shape, the required cutting length according to the shape of the initial material will be different, but in the end, the total cutting volume due to cutting is almost the same. This indicates that the increased thickness and loss volume of initial billets can be controlled by the mechanical properties and cutting shape of initial billets for targeted bi-metal curved pipes.

## 4. Discussion

Considering the relatively larger value of LR, it is inferred that the formability of the inner pipe with relatively lower strength influenced the dimensions of the outer pipe owing to the fast-forming start time of the inner section. The thickness of the bent pipes can be controlled by considering the spring-back phenomenon and the force from the inner side to the outer side by modifying the design of the mandrel or assigning design conditions to the die that contacts directly with the surface of the outer pipe [20].

Forming stability can be increased by considering LR while designing the top and bottom lengths of the material and the angle of the mandrel in contact with them to allow the top and bottom surfaces of the material to contact the mandrel simultaneously. When the proper cutting angle of the initial material is fixed, a method of changing the shape of the mandrel to match the angle of the initial material can be also considered. The contact area between the punch and billets that can minimize the load and moment are also important design factors that influence the thickness of the target product. Relative to these factors, the dimensions of the formed curved pipe product subjected to forging that provided machining allowance can be corrected by performing additional face milling. When machining allowance is selected, the secondary cutting length that considers the

vertical area where the punch meets billets in the initial cutting process must also be designed simultaneously. In this study, it was additionally confirmed through forming simulation that the loss length of the manufactured product was reduced by 8.72 mm when the vertical cutting length was 16 mm. As a result, it is necessary to analyze the detailed values of the secondary cutting length considering the face milling.

In the CFJ process, cutting standard requires a precise analysis of the load, which is an important factor to be considered. Among the components of piping, the curved pipe part is relatively more vulnerable than other parts. As the straight pipe connected to a curved pipe distributes the load acting on the curved pipe when the load exceeds the yield stress, the fracture probability and forming load of the curved pipe during forming are determined by the length of the connected straight pipe [21]. However, considering the cut initial billets, there is no straight pipe to distribute the load. Therefore, it is necessary to examine the design conditions of initial billets and dies that can reduce the forming load. Development of other additional methods can also be considered to improve the product quality. In a study case in which the bi-metal free bending method without an external die that serves to set the forming direction was applied to the bi-metal billets with inner billets made of copper and an outer thing made of aluminum, the thickness of the outer curved pipe decreased and the thickness of inner curved pipe increased. Therefore, it can be seen that there is a difference in thickness change depending on the forming method and materials [22]. It is judged that the criteria for securing the dimensional accuracy regarding the thickness of bi-metal curved pipes can and should be applied differently according to the difference in the method of each designed process.

## 5. Conclusions

In this study, reviews related to the bending process of billets using bi-metal were performed to secure precise dimensions in the manufacture of short radius elbow. The results of the study should be summarized as follows:

(1) It was verified that it is possible to secure the dimensional precision of the prototype by controlling design variables and adopting the bending method of lined pipes via the CFJP proposed on the production of a bi-metal curved pipe. The minimum top and bottom lengths of the initial material can be obtained, and the movement distance of the punch can be designated using the mathematical calculation of the center-to-end length and outside diameter of the bi-metal curved pipe. Considering the loss rate by cutting and surface machining after forming, the CFJ process design was realized to secure the dimensional precision of the target bi-metal bent pipe.

(2) During the forming of the bi-metal curved pipe composed of the materials specified in this study, the thicknesses of both the outer and inner pipes tended to increase more than the thickness of the initial material. Regarding the thickness of the inner pipe, which exhibits higher thickness sensitivity than the outer pipe, it was verified that it is possible to secure dimensional precision by setting top and bottom tolerances using load control and arrangement based on the bonding force between the outer and inner pipes, including the geometry design of the mandrel.

(3) The forging experiment and forming simulation results verified that the setting of the top and bottom lengths of the initial material influences the material filling rate in the die during forming, including subsequent changes in outer and inner pipe thicknesses and the overall length. For producing a bi-metal bent pipe comprising an outer pipe made of API 5L X65 and an inner pipe made of DSS which complies with the SR dimensions of the DN 80 standard, it is considered appropriate to employ an initial material designed to have a top length of 200 mm and a bottom length of 60–70 mm. As a result, the vertical cutting length is 16 mm for the secondary cutting process of an initial material with a top length of 200 mm and a bottom length of 60 mm. The first required cutting length is 5.69 mm, and the second required cutting length is 1.77 mm for analytical values, which means that the desired machining allowance can be controlled by secondary cutting for additional surface processing.

**Author Contributions:** Project administration, J.P.; supervision, S.K. and J.P.; formal analysis, S.H. and D.C.; literature review, J.P., S.K., S.H. and D.C. All authors have read and agreed to the published version of the manuscript.

**Funding:** This research was funded by the Korea Institute for Advancement of Technology (KIAT) grant funded by the Korean Government (MOTIE) (P0002092, The Competency Development Program for Industry Specialist).

**Institutional Review Board Statement:** Not applicable.

**Informed Consent Statement:** Not applicable.

**Data Availability Statement:** Not applicable.

**Conflicts of Interest:** The authors declare no conflict of interest.

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
