# Peer review of "A Study on the Design of Bending Process According to the Shape of Initial Billets for Bi-Metal Elbow"

_metals, doi:10.3390/met12091474_

Round 1
Reviewer 1 Report (Previous Reviewer 1)
1. In fig.1, the unit of curvature radius shall be mm. The curvature is constant。
2.“This study proposes the cut-forging-joint process (CFJP)” the process is not new. See literature “ Optimization on blank and technology parameters for elbow cold pushing process, Xue YANG, Yanshan University, 2013”
3. The review of elbow forming process is not complete. "Development of a cold stamping process for forming single-welded elbows. International Journal of Advanced Manufacturing Technology, 2017, 88: 1911-1921." et al.
Author Response
Response to Reviewer 1 Comments
Point 1: In fig.1, the unit of curvature radius shall be mm. The curvature is constant。
Response 1: As you pointed out, the unit error has been corrected and reflected in Figure 1.
Point 2: “This study proposes the cut-forging-joint process (CFJP)” the process is not new. See literature “Optimization on blank and technology parameters for elbow cold pushing process, Xue YANG, Yanshan University, 2013”
Response 2:
First of all, thank you so much for introducing me to such a good case study.
We have checked the contents of that part and corrected the contents. (“ Optimization on blank and technology parameters for elbow cold pushing process, Xue YANG, Yanshan University, 2013”)
If the prior research was conducted with the goal of optimal design, this study believes that there is a difference in forming a double curved pipe at the same time by forming simulation. Although there may not be much difference in the research mechanism, I hope that you will look at the meaning from a new trial and review level.
Point 3: The review of elbow forming process is not complete. "Development of a cold stamping process for forming single-welded elbows. International Journal of Advanced Manufacturing Technology, 2017, 88: 1911-1921." et al.
Response 3:
This study focused on precise dimensions and the amount of discarded material. If necessary, data such as forming load can be additionally entered, but this range may lead to too broad content at the core of this paper, so it was decided to briefly mention the occurrence figures in writing. However, in response to your advice, we have added the document to our bibliography for reference by other readers.
‘Gaochao, Y. Development of a cold stamping process for forming single-welded elbows. International Journal of Advanced Manufac-turing Technology, International Journal of Advanced Manufacturing Technology. 2017, 88, 1911-1921.’
And, by referring to the relevant literature, the symbols required for analysis are written in the graph analyzing the thickness and the amount of discarded material in Figure 6.

Reviewer 2 Report (New Reviewer)
Please see attached.

Author Response
Response to Reviewer 2 Comments
Point 1: Fig. 2 needs a bit more explanation. What is the process principle? Also, add labels in the Figure, eg, punch, mandrel, die, billet, etc.
Response 1:
The content of the process has been reviewed and errors have been corrected in the content of the text. And I added the content of the CFJP process to the introduction. Figure. 2, the names are indicated according to the advice.
Point 2: It is a bit confused about the dimensions involved in the initial billet and the targeted curved pipe. What’s their relationship? Or how to determine the initial billet dimension (top and bottom length, cutting angle) for a given curved pipe dimension? Since all parameters are already symbolised in Table 2, it is suggested to use these symbols to explain the relationship between billet and curved pipe. Give some quantitative relationships in equations, rather than just describing in text which is quite lengthy and not clear. Besides, there is no point in listing these symbols if they are not used at all.
Response 2:
To avoid confusion about the meaning of each symbol, Figure. 6 is indicated by the relevant dimension notation. Alternatively, the initial billet dimensions (top and bottom lengths, cut angle) for a given curved pipe dimension are the criteria set by dividing the independent variable and the dependent variable in some cases. spacing based on.
Point 3: In the Introduction, to increase readers’ interest, please add a bit more refs about applications of curved pipes in the very beginning, e.g: ‘Advances and Trends in Forming Curved Extrusion Profiles’, ‘A Review on the Potential Applications of Curved Geometries in Process Industry’. Except for the rotary draw bending process to fabricate curved pipes, other recently developed methods should be briefly introduced, such as the direct extrusion-bending forming process, please refer to: Int. J. Mach. Tools Manuf. 43 (2003) 1571-1578; Int. J. Mach. Tools Manuf. 126 (2018) 27-43; Metals 12 (2022) 877
Response 3:
In accordance with the advice, the contents of the recently carried out new process were described and references were indicated.
Point 4: Add all the symbols in Fig. 3, along with the specific dimensional values. eg, ‘Lt=200’
Response 4:
The part you mentioned has been corrected and shown in Figure 3.
Figure 3. Geometry of initial billet and die: (a) Target short radius elbow; (b) Designed initial billet.
Point 5: Fig.4, the top figure is too large, while the bottom three figures are too small.
Response 5:
The part you mentioned has been corrected and shown in Figure 4.
Figure 4. Fabrication process of short radius elbow.
Point 6: Line 260, depending ob?
Response 6:
The part you mentioned has been corrected.
(depending ob -> depending ob)
Point 7: Fig. 5a-b is missing. The front size in fig. 6 is too large.
Response 7:
I added the missing picture.
(Figure. 5(a, b))
Figure 5. Predicted shape of elbow using forging analysis contained experiments.
Modified Figure 6 by adding a picture showing the symbol and the metric. If it is judged that the previous version of the graph is better, I will reduce the figure as a whole and modify it again.
Point 8: Add symbols which represent length ratio in Fig. 6, i.e. which ratio?
Response 8:
As you said, Figure. 6 shows the symbol mark of LR.
Point 9: The conclusions section is way too lengthy, make it more concise.
Response 9:
Based on your advice, unnecessary content in conclusion (5) has been deleted and summarized into three conclusions.

Round 2
Reviewer 1 Report (Previous Reviewer 1)
I think the manuscript of the current version can be published.
Author Response
Response to Reviewer 1 Comments
Point 1: I think the manuscript of the current version can be published. In fig.1, the unit of curvature radius shall be mm. The curvature is constant。
Response 1: Thank you very much for all the kindness you have helped with the detailed review and comments.

Reviewer 2 Report (New Reviewer)
Many thanks for the response and revison. In the future, I hope the authors can mark the revised content in a different colour, this is commonly used/required for all journal papers. Few minor revisions are needed: Use the surname when citing literatures, line 66 should use 'Zhou et al. ' which is the surname. Also, revise the refs list 10-11 where the surname should be kept, and given name should be abbreviated, currently it is the opposite.
Author Response
Response to Reviewer 2 Comments
Point 1: Many thanks for the response and revison. In the future, I hope the authors can mark the revised content in a different colour, this is commonly used/required for all journal papers. Few minor revisions are needed: Use the surname when citing literatures, line 66 should use 'Zhou et al. ' which is the surname. Also, revise the refs list 10-11 where the surname should be kept, and given name should be abbreviated, currently it is the opposite.
Response 1:
Regarding the additional references you mentioned all parts have been corrected.
Thank you very much for all the kindness you have helped with the detailed review and comments.

This manuscript is a resubmission of an earlier submission. The following is a list of the peer review reports and author responses from that submission.
Round 1
Reviewer 1 Report
1.The literature on elbow forming methods is insufficient.
2. About the friction coefficient, 0.9 bi-metal and 0.02 dies is set. Why and on what basis?
3. In Figure 4, the mechanical properties of the two metals are different with different strain rates. Which curve is used in table 3, and which curve or all curves are used for simulation?
4. There is no clear experimental formed elbow and elbow section.
5.The process optimization design is not reflected in this paper. The title claims to optimize the design.
Author Response
Before answering, thank you for your valuable time and experience.
Point 1: The literature on elbow forming methods is insufficient.
Response 1:
In essence, we focused on presenting a method that can compensate for the shortcomings compared to RBD, which is the most widely used in lined pipe manufacturing.
We reviewed the missing parts of the elbow forming method, and supplemented the contents of References [4-8].
Point 2: About the friction coefficient, 0.9 bi-metal and 0.02 dies is set. Why and on what basis?
Response 2:
In order to minimize the influence of the formability due to the friction coefficient in the analysis simulation, it is assumed that the bi-metal billets are joined with the friction conditions as 0.9.
Friction between dies (outer die, mandrel) is almost non-existent, therefore the coefficient of friction between outer die and mandrel is set to 0.02 like the sliding conditions.
The coefficient of friction between punch and billets is set to 0.2.
Point 3: In Figure 4, the mechanical properties of the two metals are different with different strain rates. Which curve is used in table 3, and which curve or all curves are used for simulation?
Response 3:
The graph shown on the right of the Figure 4 expresses the flow curve of API 5L X65 and duplex stainless steel. The flow curve is obtained through a tensile test, and the load and stress required for deformation increase according to the strain rate at which the material is deformed. In a tensile test, the rate of strain generated versus static load is proportional to the load, and it is a normally more important factor especially in hot forming than cold forming. However, in this study, cold forming was dealt with, and since the forging speed of the punch, the yield strength of the material, the shape of the initial material, the generated load, the effective strain, etc. were set as variables, the important factors affecting the result Flow curve with velocity is indicated according to the strain rate and materials.
The mechanical properties in Table 3 are the values ​​interpolated using Hollomon Equation to predict the deformation properties after the 28% and 35% elongation, which are the limits of the nominal stress-nominal strain curve obtained from the tensile test.
Point 4: There is no clear experimental formed elbow and elbow section.
Response 4: At the time of the research, we couldn’t obtain a photo of the prototype under the circumstances, it was not possible to attach a photo of the prototypes in this paper.
Point 5: The process optimization design is not reflected in this paper. The title claims to optimize the design.
Response 5: After careful consideration, the title has been changed as follows.
"A study on the design of the bending process according to the shape of initial billets for bi-metal elbow"
Acknowledging that the previously used title is too broad an expression, it has been modified to fit the goal of this paper.

Reviewer 2 Report
(1) The thesis title does not match the research content. The paper only studies one process parameter, not the optimal design. In addition, the subject of the research - elbow is not reflected in the title, and the " Bending Process Using Bi-Metal Pipes" in the title is very easy to mislead readers.
(2) The process studied in this paper is a common mature process in the forming of single metal elbow. There is no innovation in process. The particularity of the research object bi-metal pipe has not been fully demonstrated in the paper. What are the special problems of pushing short-radius elbows with bi-metal pipe, and how to solve these problems, this paper has not carried out research.
(3) The title of the thesis is called Optimal Design, but the goal of the optimal design is not described in detail in the paper.
In summary, the manuscript is not innovative enough and does not meet the publication standards of the journal, and it is recommended to reject the manuscript.
Author Response
Before answering, thank you for your valuable time and experience.
Point 1: The thesis title does not match the research content. The paper only studies one process parameter, not the optimal design. In addition, the subject of the research - elbow is not reflected in the title, and the " Bending Process Using Bi-Metal Pipes" in the title is very easy to mislead readers.
Response 1:
After careful consideration, the title has been changed as follows. "A study on the design of the bending process according to the shape of initial billets for bi-metal elbow"
Acknowledging that the previously used title is too broad an expression, it has been modified to fit the goal of this paper.
Point 2: The process studied in this paper is a common mature process in the forming of single metal elbow. There is no innovation in process. The particularity of the research object bi-metal pipe has not been fully demonstrated in the paper. What are the special problems of pushing short-radius elbows with bi-metal pipe, and how to solve these problems, this paper has not carried out research.
Response 2:
The study was conducted on the difference in formability between single and double metal elbow shown in Figure 6 (a, b). In this paper, the main point we hope to emphasize is not the specificity of the double elbow, but the forming method of the targeted double elbow. In the Results and Discussion parts, we discuss the special problems (load, punch travel distance, friction force according to the tolerance of the mandrel, the amount of initial material discarded, etc.) when pushing a short-radius elbow with a bi-metal pipe and how to solve them.
However, if there are any shortcomings, we will do our best to supplement them.
Point 3: The title of the thesis is called Optimal Design, but the goal of the optimal design is not described in detail in the paper.
In summary, the manuscript is not innovative enough and does not meet the publication standards of the journal, and it is recommended to reject the manuscript.
Response 3:
After careful consideration, the title has been changed as follows. "A study on the design of the bending process according to the shape of initial billets for bi-metal elbow". Acknowledging that the previously used title is too broad an expression, it has been modified to fit the goal of this paper.
Absolutely thank you for reviewing the direction of the paper again.

Reviewer 3 Report
This is an article where the authors optimize the design of bending process of bi-metal pipes. The article is written at a good scientific and technical level, and its results are of both practical and scientific interest. The authors confirm the adequacy of the results and the efficiency of the ideas by the good agreement between the FEM calculations and experiments. The article can be published after addressing the following issues:
1. The introduction should be formulated more specifically. For example, if Fig.1 has not yet been published, then it should be moved to "2. Materials and Methods", and if it has already been published, then just a reference is needed. It is also necessary to more specifically formulate the main problem and main aim of the work.
2. In fig. 6 a colorbar does not sign? Therefore, it is not clear what is shown there.
3. Conclusions must be written more specifically. For readers, the most important thing is the scientific results obtained by the authors. The practical results are also certainly of interest, but they are not needed in the conclusions.
4. The reviewer recommends adding a scheme or table and to indicate all the process parameters that are involved in the optimization, as well as the criteria by which the optimal result is selected.
5. English language needs to be corrected.
Author Response
Before answering, thank you for your valuable time and experience.
Point 1: This is an article where the authors optimize the design of bending process of bi-metal pipes. The article is written at a good scientific and technical level, and its results are of both practical and scientific interest. The authors confirm the adequacy of the results and the efficiency of the ideas by the good agreement between the FEM calculations and experiments. The article can be published after addressing the following issues:
- The introduction should be formulated more specifically. For example, if Fig.1 has not yet been published, then it should be moved to "2. Materials and Methods", and if it has already been published, then just a reference is needed. It is also necessary to more specifically formulate the main problem and main aim of the work.
Response 1:
We move the Fig. 1 to "2. Materials and Methods" as per your advice. The contents of the presentation of the CFJ Process and its specific goals have been supplemented.
Point 2: In fig. 6 a colorbar does not sign? Therefore, it is not clear what is shown there.
Response 2:
I have edited the picture by supplementing the part you mentioned in Figure 6.
Point 3: Conclusions must be written more specifically. For readers, the most important thing is the scientific results obtained by the authors. The practical results are also certainly of interest, but they are not needed in the conclusions.
Response 3:
I have merged 3 and 4 of the conclusion part, and we will add what is needed to the conclusion.
Point 4: The reviewer recommends adding a scheme or table and to indicate all the process parameters that are involved in the optimization, as well as the criteria by which the optimal result is selected.
Response 4:
After careful consideration, the title has been changed as follows. "A study on the design of the bending process according to the shape of initial billets for bi-metal elbow"
Acknowledging that the previously used title is too broad an expression, it has been modified to fit the goal of this paper and we would consider to add the contents of process parameters.
Point 5: English language needs to be corrected.
Response 5:
We will do our best to correct it. Sincerely thank you.

Round 2
Reviewer 1 Report
The research is not extensive and in-depth. The coefficient of friction has not been verified. In this process, the strain rate cannot reach the range in Figure 4. It is incomprehensible that the pictures of experimental forming elbow cannot be provided. The optimal design of blank in this process has been studied in relevant literature. ( Xue Yang. Optimization on blank and technology parameters for elbow cold pushing process. 2013)
Reviewer 2 Report
The article does not show what changes have been made. In addition, the process proposed in the paper is a mature existing process.
Reviewer 3 Report
I read the article, and I believe that the Authors have improved it, and now it is ready for publication.